# Direct Method of Measuring the pH Value of Wood

**Anton Geffert \*, Jarmila Geffertova and Michal Dudiak**

Faculty of Wood Sciences and Technology, Technical University in Zvolen, T.G.Masaryka 24, 96001 Zvolen, Slovakia; geffertova@tuzvo.sk (J.G.); miso.dudiak@gmail.com (M.D.)

\* Correspondence: geffert@tuzvo.sk; Tel.: +421-45-520-6378

**Abstract:** A direct method of measuring the pH value of wood substance is proposed in the paper. The achieved results were completed by determining the pH value on the wood surface using the contact method. Moreover, the results were compared to the results achieved using the indirect methods to determine the pH value in cold water, as well as hot water, extract of wood. Using the direct method for measuring the pH value in drilled fresh sawdust, the pH value of beech was 5.11, of birch was 5.29, of alder was 4.88, and of maple was 4.65. Following the achieved results, the possibility to measure the pH value using a fast, accurate method useful in practice complying with the condition of the minimum free water in wood (moisture content of wood above the fibre saturation point) was presented. The results of measurements of the pH value using the contact method on the wood surface showed that this method can be used in the case of coniferous as well as broadleaved trees with heartwood. The value of pH measured on the surface of pine was 4.50, of spruce was 4.79, of the heartwood of oak was 3.46, and of the sapwood of oak 5.04. The measurement of pH values of water extracts confirmed great dependence of measured values on the conditions of wood extraction.

**Keywords:** pH value of wood; flat electrode; piercing electrode; water extraction of wood; beech; spruce; pine; oak; birch; alder; maple

## 1. Introduction

The pH value is a measure of the concentration of $H^+$ ions in a solution and it is used to determine acidic, neutral, or base behaviour [1].

pH values are very important physiological parameter for plants, humans, and animals. The measurement of the pH value is used in almost every laboratory, manufacturing plant, or field-work [2].

All wood species have a pH and most species are naturally acidic, with the vast majority having a pH between 4.0 and 5.5. The values of pH of various tree species of temperate zone range from 3.3 to 6.4. Tropical tree species are of a weak acid or a weak base ranging from 3.7 to 8.2. Of course, there are examples of strongly acidic species like Douglas fir, with the pH value of 3.3, and alkaline species such as African tree species (*Dalbergia melanoxylon* Guill.; *Terminalia superba* Engl. & Diels), with the pH value of 8.0 and 8.2 [3–6].

The acidic reaction of most tree species is caused by free acids and acidic groups easy to be separated, that is, especially by acetic acid and acetyl groups. The acidity of wood is increased as a result of the wet environment and higher temperature [1].

Knowing the pH value of wood is very important in various areas of its use. Metals can corrode or the performance of adhesives, as well as the fixation mechanism of wood preservatives, can be affected when coming into contact with wood. The pH value of wood must be the centre of our attention, as well as in the case of producing the wood pulp, fibreboards, and particleboards or plasticising wood [3,6–10].

The value of pH is a fast and easy to measure parameter; therefore, it is used to control and manage technology and processes.

Several other methods of measuring the pH value of wood have been developed, as there are uncertainties facing the direct methods for measuring solid timber when the moisture content of wood is above the fibre saturation point of wood [1].

The pH value of wood is measured using following methods, which requiring more time and handling:

- The way of determining the pH value when the pH value of acidic or base solution is modified to the pH value of disintegrated wood that is suspended [3].
- The method of slope adjustment, graphical evaluation of the pH value of wood following the measurement after it is immersed in diluted NaOH and the solution of HCl [4,11].
- A more direct method of measuring the pH value is based on measuring the press-extracted wood fluids [12].

The absolute pH value of wood is not provided by the often used cold water and hot water extraction of wood by a certain amount of water [13–16]; thus, it can be used only to compare the determined values. The measurement of wood extractions is distorted as wood also contains acid insoluble groups in cell wall polymers [6].

The aim of the paper was to verify the possibility of determining the pH value using the contact method on the wood surface, to propose a fast and easy to be applied method in practice for measuring the pH value of wood substance at the different moisture content. Moreover, the achieved results were compared to the pH values achieved using the indirect methods for determining the pH value in cold water, as well as hot water, extraction of wood.

## 2. Materials and Methods

### 2.1. Material

Samples of air-dried and green wood of the widespread broadleaved and coniferous tree species, beech, oak, spruce, and pine, were used in the experimental measurements of the pH value of wood by the flat electrode on the wood surface.

Samples of the broadleaved tree species: green wood of beech ($w_a$ = 68.5%), birch ($w_a$ = 75.9%), alder ($w_a$ = 71.8%), and maple ($w_a$ = 60.8%) were used for direct measuring of the pH value of the wood substance.

### 2.2. Methods

The samples of the wood trees were defined in terms of chemical composition: Extractives ASTM D 1107-96. Standard Test Method for Ethanol-Toluene Solubilityof Wood [17]

| | |
|---|---|
| Polysaccharide fraction | Chlorite isolation method of Wise [14] |
| Cellulose | Kürschner-Hoffer method [14] |
| Lignin | ASTM D 1106-96. Standard Test Method for Acid Insoluble Lignin in Wood [18]. |

The fraction of sawdust from 0.5 mm to 1.0 mm from completely disintegrated boards of the wood trees (including surface and center part) was used to monitor the chemical characteristics.

Contact measurement on the wood surface using the contact combination electrode with glass shaft SenTix Sur and inoLab pH metre was the first mentioned method for determining the pH value of wood. The method was applied in accordance with the standard STN 50 0374 testing of pulp and paper [19]. Surface pH measurement of paper and pulp. Testing is based on the measurement of the pH value of the sample surface moistened with one drop of water using the planar combination electrode.

In the developed method of direct determination of the pH value of wood, pH meter SI 600 with a piercing probe LanceFET+H from the company SENTRON (steel tipped probe for penetration

*2270-010* and measurement of non-liquids) was used. The moisture content of samples of green wood of investigated tree species with the dimensions of 600 × 90 × 32 mm was above the fibre saturation point. Three measurements of each sample were carried out in the centre, and 10 cm away from the edge.

Sawdust was drilled using an electric drill machine, pressed back into the opening, and subsequently, a piercing electrode was applied. After achieving the appropriate contact between sawdust and the probe, changes in the pH value were recorded every 15 s for a seven-min period.

The pH meter Checker by Hanna and wood samples of investigated tree species in the form of dried sawdust of two qualities (fine-grained fraction with 0.5–1.0 mm$^2$ and fraction consisting of sawdust with more than 1.0 mm$^2$) were used in comparative indirect measurements of the pH value of wood extracts.

When measuring the pH value in the cold water extract, 2.0 g of sawdust was mixed with 40 cm$^3$ of distilled water in a beaker. After 24-h contact of sawdust with water at the laboratory temperature, the pH value of cold water extract was measured.

When measuring the pH value in the hot water extract, 2.0 g of sawdust with 40 cm$^3$ of distilled water was used again. Sawdust was extracted in the water bath at the temperature of 95 °C for 15 min. After cooling, the pH value of extract was measured.

The measurement progress of the pH value of selected tree species was processed in the program STATISTICA 12. The one-way analysis of variance (ANOVA) program was used for the evaluation of measured data.

## 3. Results and Discussion

Basic chemical analysis of wood of the examined wood trees: extractive composition in the ethanol-toluene solvent (TEE), holocellulose, cellulose, and lignin content was conducted in order to characterise the sample. Hemicellulose content was calculated as the difference between the holocellulose and cellulose content (Table 1).

**Table 1.** Chemical composition of wood of the examined wood trees.

| Sample of Wood | TEE (%) | Holocellulose (%) | Cellulose (%) | Hemicellulose (%) | Lignin (%) |
|---|---|---|---|---|---|
| Beech | 2.3 | 82.5 | 46.7 | 35.8 | 20.7 |
| Birch | 2.9 | 84.2 | 45.4 | 38.8 | 17.7 |
| Alder | 6.6 | 77.2 | 44.1 | 33.1 | 22.0 |
| Maple | 2.5 | 80.1 | 44.6 | 35.5 | 24.9 |
| Spruce | 1.0 | 77.8 | 50.0 | 27.8 | 26.5 |
| Pine | 5.2 | 73.1 | 47.3 | 25.8 | 25.6 |
| Oak | 9.4 | 69.4 | 39.1 | 30.3 | 22.8 |

TEE, ethanol-toluene solvent.

The measurement on the wood surface using the contact combination electrode SenTix Sur was the first analysed method for determining the pH value of wood (Figure 1).

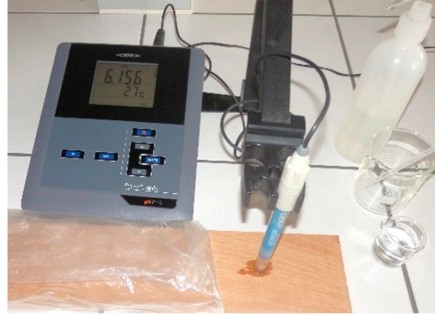

**Figure 1.** Measuring the pH value of wood using planar combination electrode.

When measuring the air dried sample of beech wood ($w_a$ = 7.5%), the pH value could not be measured as the drop of water was absorbed by the wood structure very quickly and the value of pH was not stabilized. Fast penetration of liquids into beech wood relates to the chemical composition and capillary-porous structure, especially to high proportion of vessels and medullary rays [20], causing its good permeability. Reinprecht [21] mentioned that broadleaved tree species, beech, poplar, hornbeam, maple, and birch (but not the heartwood of oak and locust tree), are more permeable in comparison with coniferous trees.

When measuring the pH value using the contact method on the surface of air-dried wood of pine and spruce, the value of pH stabilized very quickly. The average value of pH of pine wood was 4.50 ± 0.04 and in the case of spruce wood, the measured pH value was 4.79 ± 0.11 (Figure 2).

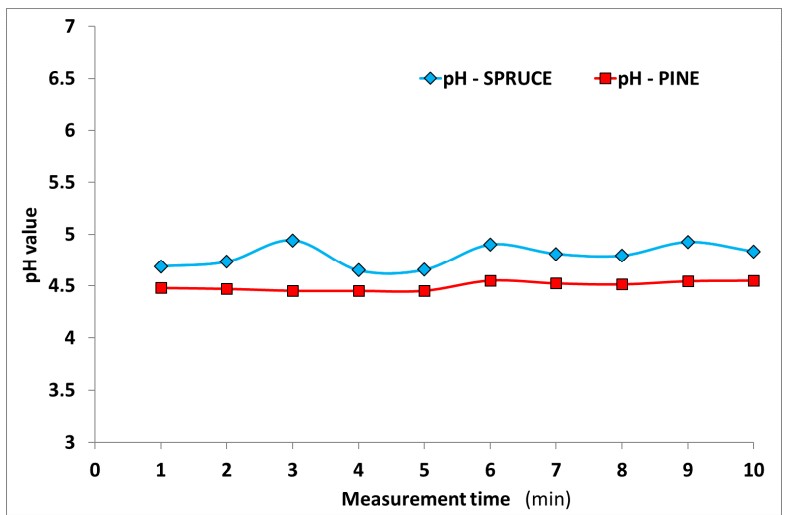

**Figure 2.** The pH value of the wood surface—spruce and pine.

The results of measuring the pH value achieved by contact method on the surface of dry wood indicated that this method is suitable only in the case of tree species with a high content of resin and fatty acids such as coniferous trees. High resin acid content prevents the drop of distilled water from penetrating the structure. Assarssson and Akerlund [22] mention that the resin acid content in the petroleum ether extract of spruce is 27.37% and of fir is 29.03%.

When measuring the pH value of the green beech wood sample using the flat electrode, the pH value stabilized for a long time. After repeated contact of the flat electrode with the wood sample taking 10 min, the measured values of pH ranged from 5.01 to 6.56.

Contact method was also used to measure the pH values of heartwood and sapwood of the fresh sample of English oak with the moisture content above the fibre saturation point characterised by a higher content of resin and fatty acids [23–27]. Average values of pH determined after five-min contact of the flat electrode with the wood sample were 3.46 in the case of heartwood and 5.04 in the case of sapwood.

According to the standard STN 50 0374 [19], the pH value is measured using the flat electrode only on the surface of wood moistened with a drop of water. The results of measurements indicated the fact that this method cannot be used to measure the pH values of broadleaved trees with low moisture content and low content of resin and fatty acids.

When proposing the direct method of measuring the pH value of wood in the mass, the selection of the right electrode played an important role. Owing to its robust construction, piercing electrode LanceFET+H from the company SENTRON applied to partially disintegrated beech wood was the most suitable for the experiment. Measuring the pH value was carried out in the drilled sawdust pressed in the container of an appropriate size (Figure 3).

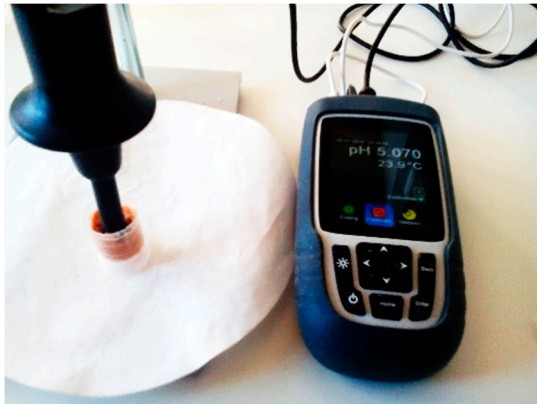

**Figure 3.** Measuring the pH value using the piercing electrode in sawdust.

The moisture content of wood above the fibre saturation point is an essential condition of the proposed direct method. Therefore, the method was verified by measuring the pH value using the sawdust moistened by distilled water in various proportions (Figure 4).

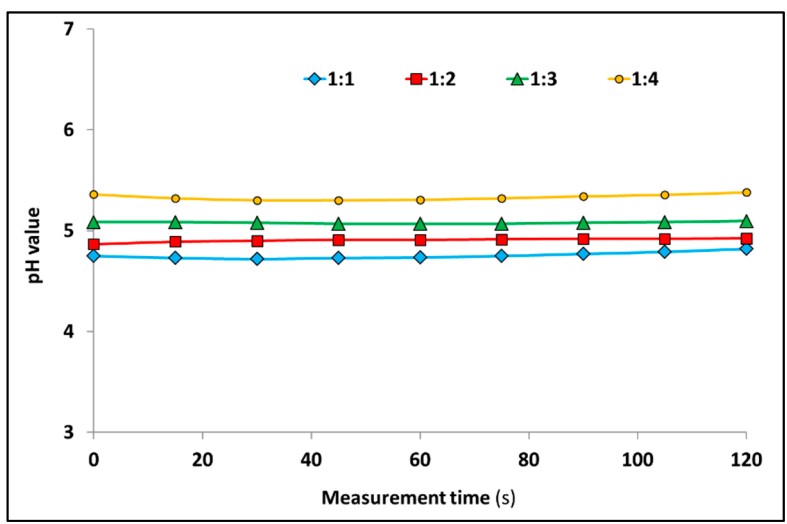

**Figure 4.** The pH values of wood with different moisture contents.

Measured pH values were stabilized very quickly within a few tens of seconds. The pH value was increasing, that is, the concentration of $H_3O^+$ ions (acidity) as well as the amount of substance of $H_3O^+$ was decreasing with the increased quantity of added water. Calculating the concentration of $H_3O^+$ ions and, subsequently, the amount of substance of $H_3O^+$ was based on the formula for the pH value for weak acids and the acidity was considered acetic acid with $pK_a = 4.756$ (Table 2).

**Table 2.** The pH value of sawdust moistened with different quantity of distilled water.

| Ratio Wood/Water | pH | $H^+ \times 10^5$ (mol/L) | $n_{H^+} \times 10^8$ (mol) |
|---|---|---|---|
| 1:1 | 4.73 | 1.977 | 1.977 |
| 1:2 | 4.91 | 0.893 | 1.726 |
| 1:3 | 5.07 | 0.413 | 1.239 |
| 1:4 | 5.30 | 0.143 | 0.573 |

Changes in parameters in Table 2 indicate that final acidity is significantly affected not only by the released acetic acid, but also by the acid components in wood cell walls.

The measurement results of the pH values of moistened sawdust indicated that the pH value of wood with higher moisture content can be measured with accuracy and speed using this method.

In the following part, the results of direct measurement of the pH value of green wood of selected broadleaved tree species (beech, birch, alder, and maple) are presented. Measuring the pH value was carried out by piercing electrode in drilled sawdust pressed back into the drilled opening in three various parts of the sample. In all cases of tree species, the measured pH values were stabilized after 45 to 60 s of contact of the electrode with the sawdust.

The measurement progress of the pH value of beech, birch, alder, and maple is shown in Figure 5. The observed differences between the mean values of individual tree species were not assessed as random errors, but as statistically significant.

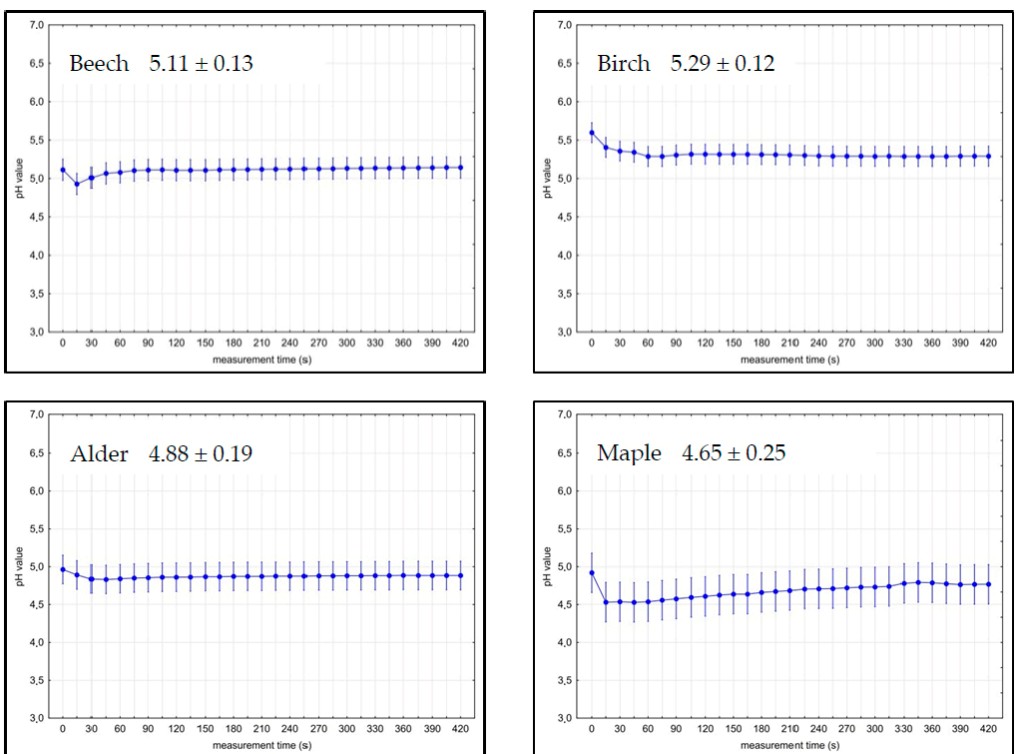

**Figure 5.** Graph of 95% confidence intervals of the pH value of green wood—beech, birch, alder, and maple.

Average pH values of green wood of selected broadleaved tree species at the defined moisture content $w_a$ are illustrated in Table 3 in descending order.

**Table 3.** Average values of the pH of green wood.

| Tree Species | Birch ($w_a$ = 75.9%) | Beech ($w_a$ = 68.5%) | Alder ($w_a$ = 71.8%) | Maple ($w_a$ = 60.8%) |
|---|---|---|---|---|
| pH | 5.29 ± 0.12 | 5.11 ± 0.13 | 4.88 ± 0.19 | 4.65 ± 0.25 |

When the sample of green beech wood was used for the time ranging from 60 to 420 s, the pH value increased linearly with the narrow range of 5.11 ± 0.13. The pH values measured in three various parts of the sample were almost identical as well. After the stabilized stage with the average pH value of green beech wood of 5.29 ± 0.12, a small pH range occurred. In the case of the sample of alder wood, the average pH value of 4.88 ± 0.19 was stabilized. The pH values of maple wood were in the observed range of 4.65 ± 0.25. In the case of the third measurement of the pH value of maple wood with the

highest values of pH, the pH stability was the lowest owing to the change in moisture content resulting from the water evaporation and poor contact of sawdust with the electrode.

Cold water and hot water extraction was performed using the sawdust of two different qualities (sawdust fraction of 0.5–1.0 mm$^2$ and coarse fraction of sawdust) of selected tree species. The pH value of achieved extracts was measured using the pH meter Checker (Figure 6).

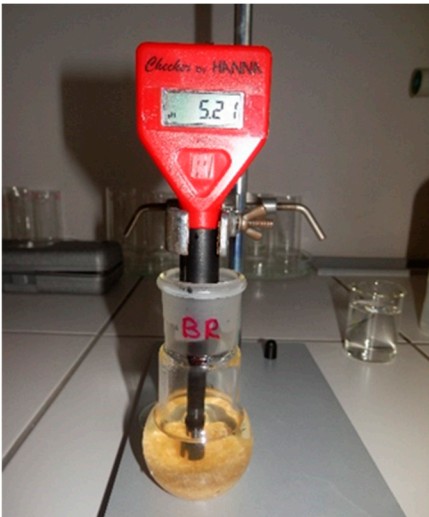

**Figure 6.** Measurement of the pH value using Checker in water extracts from sawdust.

Measuring the pH value of wood extracts is often used as the method in practice. However, it is time consuming and the information associated with the real acidity of wood is not provided. The measurement is affected but many factors, especially extraction conditions—time, temperature, and hydro module.

Therefore, the effect of selected factors and measured results were verified by comparative measurements and the measured data are summarised in Figure 7.

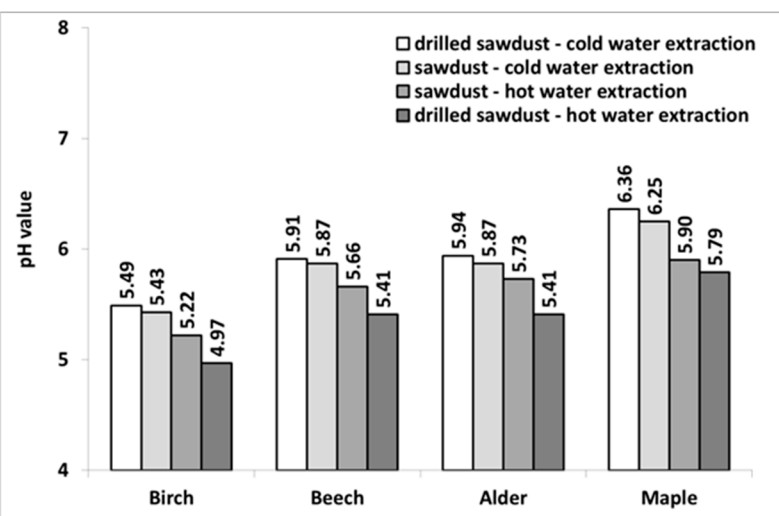

**Figure 7.** The pH value of cold water and hot water extractions.

When the effect of fraction size was assessed, the difference in the pH values of extracts after the cold water extraction taking 24 h between both fractions was in the range from 0.04 for beech wood to 0.11 for maple wood. The findings of Sitholé [15], that the pH values are not affected significantly by the sawdust fraction, were confirmed.

After hot water extraction of sawdust in distilled water at the temperature of 95 °C for 15 min, the measured pH values ranged from 4.97 for birch wood to 5.90 for maple wood. Lower pH values in comparison with cold water extraction relate to the higher intensity of acetyl groups released as a result of higher temperature, despite the shorter time of extraction.

In the process of cold water as well as hot water extraction, hydrolysis of wood polysaccharides is done in the form of the autocatalysis. Its speed increases owing to acetic acid released from the acetyl groups of hemicelluloses. In later stages of hydrothermal extraction, the acidity of the environment is increased by formic acid and levulinic acid resulting from the decomposition of monosaccharides. At the same time, hemicellulose degradation as well as amorphous proportion in cellulosic sample appear [1].

Four different methods of measuring the pH value were focused especially on the beech owing to its dominant position in wood-processing industry in the Slovak Republic. The achieved results are summarised in the Table 4.

**Table 4.** Measured values of pH of beech wood.

| Method | on the Surface | in the Mass | Cold Water Extraction | Hot Water Extraction |
|---|---|---|---|---|
| | Flat Electrode | Piercing Electrode | Glass Electrode | |
| **pH** | 5.01–6.56 | 5.11 ± 0.13 | 5.87 | 5.66 |

Measuring the pH value on the surface of wood using the contact method did not work because of the long time necessary for stabilization of the pH value and large range of measured values (5.01–6.56).

During the indirect measurement of pH value using the cold water and hot water extraction, the pH value of beech wood was the highest (5.87 and 5.66). It is caused by the dilution of $H_3O^+$ ions during the wood extraction.

The fastest and the most accurate information about the acidity of measured green wood is provided by the pH value measured in the wood substance using the piercing electrode (5.11).

## 4. Conclusions

The results showed that the piercing electrode can be used for measuring pH value from sawdust with the higher moisture content—at the moisture content above the fibre saturation point. The method is usable in the laboratory as well as in operational practice for its speed and sufficient accuracy.

Measuring the pH value using the contact method on the surface of the wood sample indicated that this method is influenced not only by moisture, but also by other factors (structure and chemical composition of wood).

The measurement of pH values of water extracts confirmed great dependence of measured values on the conditions of wood extraction.

**Author Contributions:** A.G. and J.G. designed the whole study; J.G. and M.D. conducted data collection, modeling and results analysis; J.G. wrote the original draft paper; A.G. revised and edited the paper.

**Funding:** This research was funded by project APVV-17-0456 "Thermal modification of wood with water vapor for purposeful and stable change of wood colour".

**Acknowledgments:** This experimental research was carried out under the grant project APVV-17-0456 "Thermal modification of wood with water vapor for purposeful and stable change of wood colour". The contribution was also prepared within the project agency IPA TUZVO, which contributed significantly to the creation of this contribution through the project: IPA TUZVO 15/2019.

**Conflicts of Interest:** The authors declare no conflict of interest.

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
