# Peer review of "Direct Method of Measuring the pH Value of Wood"

_forests, doi:10.3390/f10100852_

Round 1
Reviewer 1 Report
This is an interesting and a mostly well written paper about measuring the pH value of various wood species. Abstract mostly ok. Lines 18-19: Was only sapwood used in coniferous wood measurements? If so, I would suggest the authors to specify this in the text. Line 20: Language checking: There was great dependence of measured values on the… Materials and Methods: Would it have been possible to use a climate chamber with controlled moisture conditions to achieve moisture equilibrium in wood samples before measurements? Then probably green wood and also heartwood would have given less variation between measurements and stabilized more easily. But that would of course demand a bit time to get the stabilized state. This paper presents pH measurements with nice stabilization curves from samples. Results: Does the variation between measurements represent only repeated measurements and not the variation between different wood samples? Were the measurements taken only from single wood samples? In a methodological paper this could be ok, but for further conclusions from pH values between different wood samples taken at different times of year and at different moisture content this experimental design would not be ok. Figure 4: Axis titles, should be corrected in English, please. Conclusions: Language should be corrected. E.g. Lines 229-231: The results showed that the piercing electrode can be used for measuring pH value from sawdust with higher moisture content. Please, rewrite conclusions in a brief way. This was only a suggested example. Line 239: H value? Should it be pH value? Line 240: Coniferous sapwood only? Please, mention in the text, if sapwood only or both sapwood and heartwood were measured. Lines 242-245: Leave it out.
Author Response
The authors thank the reviewer for the factual and concrete comments that were written with the aim of improving the quality of the reviewed article. Received comments were considered and appropriately incorporated into our article.
- lines 18-19: the pH measurement of coniferous wood was executed at several places of the board (not only measured on sapwood),
- line 20: the comment has been taken into account and the text has been modified,
- Materials and Methods: The text of this section has been amended. The authors´ focus was not to gain deeper knowledge about the pH of selected tree species, but to verify a new direct method of pH measurement and its comparison with other used methods. Humidity control of wood samples in the climate chamber would be considerate if different work objectives set.
- Results: Differences between pH measurements represent the difference of three measurements from individual wood samples according to the procedure described in the methodology. Wider use of the measured pH values would necessitate to perform measurements on large sets of carefully selected samples. However that was not the goal of the work.
- Figure 4: replaced.
- Conclusions: the reviewer's comments on this section (lines 229-231, 239, 240, 242-245) have been taken into account and the text has been modified accordingly.
Reviewer 2 Report
The manuscript ID: forests-585847 entitled “DIRECT METHOD OF MEASURING THE pH VALUE OF WOOD” by Anton Geffert, Jarmila Geffertova and Michal Dudiak, showed the use of a direct method to measure the pH values of several woods as an alternative way to indirect methods. However, in my opinion the authors should rewrite the all manuscript by supplement it. Improve the materials and methods – (it needs to mention which statistical treatment was performed) and results and discussion. Improve also the figures.
The authors should also explain the wood samples analysed because in the materials only refer to beech but in results present values from other woods (birch, alder, maple,…)
Author Response
The aim of the work was not to obtain a large set of pH values for selected tree species, but to verify a new direct method of pH measurement and its comparison with other methods used.
After considering of all comments received:
- appropriate adjustments have been made to Materials and Methods (information on samples analyzed),
- the statistical evaluation has been extended,
- the manuscript of the article (including figure 4) has been modified accordingly.
Round 2
Reviewer 2 Report
This new version of the manuscript ID forests-585847 entitled “DIRECT METHOD OF MEASURING THE pH VALUE OF WOOD” has improved significantly. In my opinion it is appropriate to be published into florests however I have some comments/suggestions:
L180-L184 –rewrite the sentence about statistical analysis and put in the 2. Material and Methods the description and program used of the statistical methods and the analysis of the data in the 3. Results and Discussion
Figure 2 and Figure 4 - remove bold of time unit (min) and (s)
Table 2 – change mol/l to mol/L
Figure 5 - reconsider put all the information of the 4 graphs only in one graph
Author Response
The authors thank the reviewer for their comments on the article being edited, which they tried to incorporate in the article.
L180-L184 – comments on this section have been taken into account and the text has been modified and divided into sections 2 and 3 Figure 2 and Figure 4 - removed bold of time unit (min) and (s) Table 2 – changed mol/l to mol/L Figure 5 - the reviewer's comment was considered and Figure 5 for clarity was left in its original condition